# Active Learning for Breast Cancer Identification

**Xinpeng Xie**
Computer Vision Institute
Shenzhen University
Shenzhen, Guangdong, China 518060
xiexinpeng2017@email.szu.edu.cn

**Yuexiang Li**
Computer Vision Institute
Shenzhen University
yuexiang.li@szu.edu.cn

**Linlin Shen**
Computer Vision Institute
Shenzhen University
llshen@szu.edu.cn

## Abstract

Breast cancer is the second most common malignancy among women and has become a major public health problem in current society. Traditional breast cancer identification requires experienced pathologists to carefully read the breast slice, which is laborious and suffers from inter-observer variations. Consequently, an automatic classification framework for breast cancer identification is worthwhile to develop. Recent years witnessed the development of deep learning technique. Increasing number of medical applications start to use deep learning to improve diagnosis accuracy. In this paper, we proposed a novel training strategy, namely reversed active learning (RAL), to train network to automatically classify breast cancer images. Our RAL is applied to the training set of a simple convolutional neural network (CNN) to remove mislabeled images. We evaluate the CNN trained with RAL on publicly available ICIAR 2018 Breast Cancer Dataset (IBCD). The experimental results show that our RAL increases the slice-based accuracy of CNN from 93.75% to 96.25%.

## 1      Introduction

Breast cancer (BC) is one of the most common cancers in the world. 12.5% women in the United States suffer from breast cancer. The number of breast cancer patients in China dramatically increases in the recent decades. BC is difficult for diagnosis as it has no typical symptoms at the early stage.  The traditional diagnosis of BC requires pathologists to manually inspect, which is laborious and time-consuming. Hence, it is necessary to develop an automatic system to assist pathologists to effectively identify breast cancer.

In recent years, deep learning has achieved great successes in natural image classification and recognition [1, 2]. The deep convolutional networks, such as DenseNet [3], and ResNet [4], are widely known and accepted by the community. As a consequence, increasing number of works tried to apply deep learning technique to the biomedical field [5-7]. In the past several years, deep convolutional network yields the state-of-the-art performances for breast cancer classification. Chougrad et al. [8] propose a Computer Assisted Diagnosis (CAD) system, based on a deep convolutional neural network (CNN) model, for classifying breast mass lesions. They also use transfer learning and fine-tuning to deal with small medical datasets. The experimental result showed that the proposed framework is an effective "second-opinion" to help the pathologists give more accurate diagnoses. Le et al. [9] presented a patch-based

1st Conference on Medical Imaging with Deep Learning (MIDL 2018), Amsterdam, The Netherlands.

CNN for whole slide tissue image classification. They separated images into small patches and combined patch-level CNN with supervised decision fusion. However, existing automatic systems are mainly designed for the binary classification task, i.e. benign/malignant, or carcinoma/non-carcinoma, few of them was developed for multi-classes cases.

In this work, a CNN model trained with a novel strategy, namely reversed active learning (RAL), is proposed for the analysis of breast cancer. The main contribution of this work is listed in the following:

- We proposed an effective training strategy, i.e. reversed active learning, for deep learning models, which can significantly boost the performance of CNN.
- Few of work paid attention to the multi-classes breast cancer identification. We proposed a CNN to address the multi-class classification problem, which achieves a competitive result on publicly available ICIAR 2018 Breast Cancer Dataset (IBCD).

## 2    Method

We proposed a reversed active learning (RAL) training strategy to train network to automatically classify breast cancer images. we applied CNN for slice-based classification of multi-class problem. When using RAL, the problem of mislabeled data can be resolved by patch selection.

### 2.1    Image Pre-processing & Data Augmentation

The IBCD contains 400 annotated H&E stain images, which can be classified to four categories: A. Normal, B. Benign, C. InSitu and D. Invasive, as shown in Figure 1. Each category has 100 images, which are labelled according to the predominant cancer type in each image. The resolution of the breast images is 2048 x 1536, which is too large for deep learning network to directly process. Hence, we slide the window with 50% overlapping over the whole image to crop patches with size 512 x 512, as shown in Figure 2. The cropping generates 2800 patches for each class. Rotation and mirror were used to increase the size of training dataset. Each patch was rotated by 90°, 180° and 270° and then reflected vertically, resulting in an augmented training set with 896, 00 images.

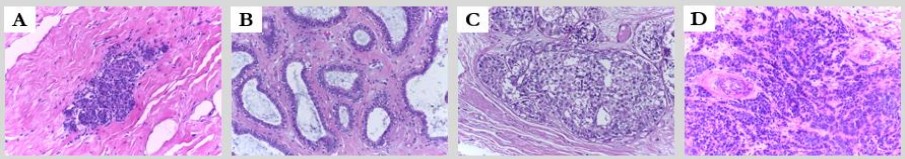

Figure 1: Examples of whole slide images in IBCD. A: Normal tissue; B: Benign abnormality; C: In Situ carcinoma; D: Invasive carcinoma.

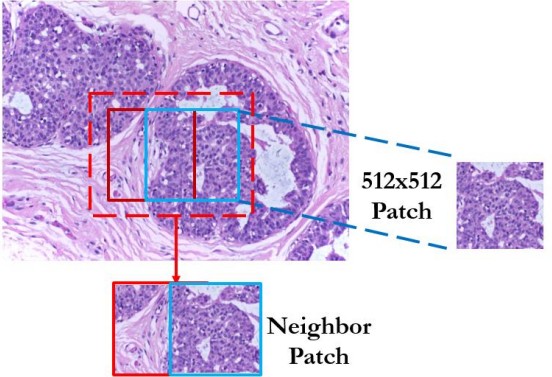

Figure 2: An example of patch (512×512) generation.

The label of whole image was assigned to the corresponding generated patches. However, some non-carcinoma areas in the breast slice may be mislabeled as carcinoma and influence

the subsequent network training. An example is shown in Figure 3. The patch contains normal tissue in Benign slice, but was labeled as Benign.

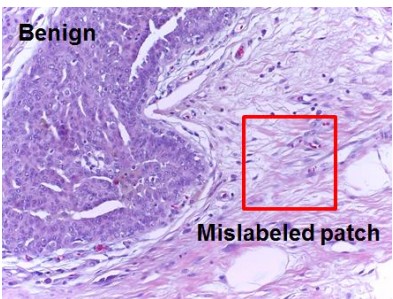

Figure 3: Example of mislabeled patch. The normal patch was labeled as Benign.

## 2.2 Reversed Active Learning (RAL)

To detect and remove the mislabeled patches, a reversed process of the active learning method proposed in [10] was proposed. Traditional active learning is proposed to train a model with incremental manually labeled data. However, in this work, we use active learning to remove mislabeled patches, which decreases the number of images of training set. Hence, it is a reversed process of traditional active learning (RAL). The flowchart of proposed RAL is shown in Figure 4.

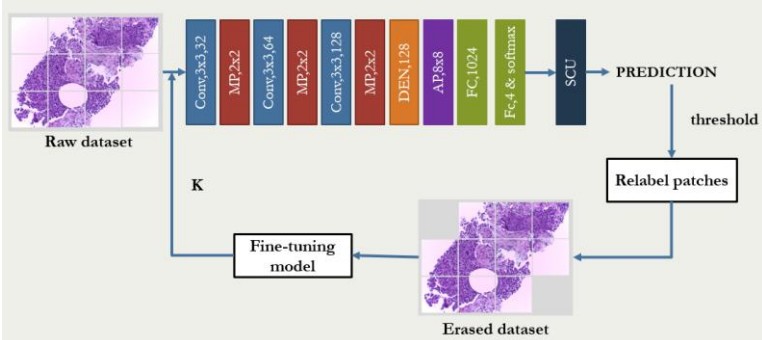

Figure 4: The pipeline of reversed active learning. K is the number of times to relabel patches.

We first trained a CNN model on original patch training set until it achieved a pre-set performance. Then, the trained model was used to make prediction on the training set. The patches with confidence lower than 0.5 were removed. Furthermore, each patch was augmented to eight patches with data augmentation ('rotation' and 'mirror'). If more than four of the augmented patches were removed, the remaining patches were removed from the training set, either. The refined dataset in Figure 3 is the one without mislabeled patches, which is then used to finetune the CNN model for a better classification performance.

After iteratively removing mislabeled patches and finetuning the CNN model, the network was continuously trained till the CNN achieved a satisfactory classification performance. Table 1 listed the number of patches after each refinement of the training.

Table 1: The numbers of training set in each iteration.

| Iteration number K | Training set |
|---|---|
| K=0 | 89,600 |
| K=1 | 89,026 |
| K=2 | 88,170 |
| K=3 | 87,363 |
| K=4 | 86,600 |

## 2.3  Network Architecture

The flowchart of our CNN model is shown in Figure 5. The blue, red, purple and green rectangles represent the convolutional layer, max pooling layer, average pooling layer and fully-connected layer, respectively. Our CNN contains 20 main layers.

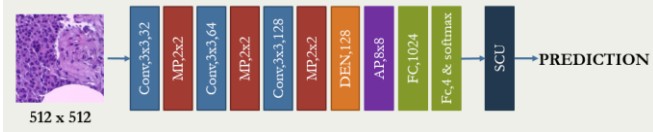

Figure 5: The architecture of proposed deep learning framework. DEN is the Deep Extraction Network module, and SCU is the Slice-base Classification Unit.

**Deep Extraction Network module (DEN).** As recent research [11] shows that a deep learning network with densely-connected architecture is more effective than a normal network. A Deep Extraction Network (DEN) module, the orange rectangle in Figure 4, is proposed to extract high-level features. The size of feature maps for deep layers are small than the shallow layers. Inspired by [12], a 1x1 convolutional layer is placed between two 3x3 convolutions for feature fusion. The architecture of DEN module is listed in Table 2. The network optimization is supervised by cross entropy loss.

Table 2: The architecture of DEN module. Pipeline consists of Convolutional layer (C) and Max pooling layer (M).

| Layer | Type | Kernel size & number |
|---|---|---|
| 1 | Convolutional | 3x3, 128 |
| 2 | Convolutional | 1x1, 128 |
| 3 | Convolutional | 3x3, 128 |
| 4 | Max pooling | 2x2 |
| 5 | Convolutional | 3x3, 256 |
| 6 | Convolutional | 1x1, 256 |
| 7 | Convolutional | 3x3, 256 |
| 8 | Max pooling | 2x2 |
| 9 | Convolutional | 3x3, 128 |
| 10 | Convolutional | 1x1, 128 |
| 11 | Convolutional | 3x3, 128 |

**Slice-based Classification Unit.** The Slice-based Classification Unit (SCU) is proposed to process whole slide tissue images. We directly divide the original image (2048 x 1536) into twelve 512 x 512 patches and calculate the class possibilities for each patch. "Major Voting" is used to yield the image-level classification. Figure 6 presents the 4 x 3 patch-based classification results. The green, blue, orange and red colors represent Normal, Benign, InSitu, Invasive, respectively.

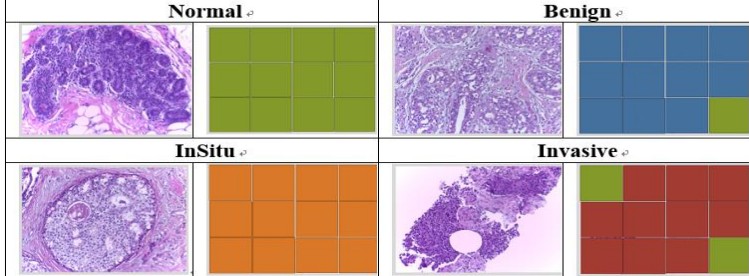

Figure 6: Classification of non-carcinoma and carcinoma breast slices. The first column is the original breast slices and the second is the corresponding maps generated by our network.

## 2.4 Implementation

The proposed framework is implemented using *Keras* toolbox. The patch-based network is trained with a mini-batch size of 64 on four GPUs (GeForce GTX TITAN X, 12GB RAM). Size of input patches is 512 x 512. The initial learning rate is set to 0.0001. 'Adam' [13] , a optimization algorithm that can replace the traditional stochastic gradient descent process, is used to iteratively update neural network weights based on training data. The network converges after 18 epochs of training.

## 3 Experimental Results

### 3.1 Dataset

The dataset employed in this study is the publicly available Breast Cancer Histology images [14] provided by ICIAR2018 Grand Challenge. We separate the dataset (400 images) to training and validation set according to a ratio of 80:20.

### 3.2 Results

We evaluate our method performances in terms of average classification accuracy (ACA) on the training set and validation set. The evaluation result is listed in Table 3. It can be observed that the proposed RAL significantly improves the patch-based classification accuracy on validation set from 89.16% to 92.81%.

With SCU, the CNN model is allowed to make slice-based prediction. As shown in Table 3, the slice-based accuracy of CNN increases from 93.75% to 96.25% using our RAL. Furthermore, we notice that after 3 times of iteratively refinement, the mislabeled patches in original training set were basically removed and the accuracy stopped increasing.

Table 3: Patch-based ACA and Slice-based ACA for different models (%).

| Model | Patch-based ACA | | Slice-based ACA | |
|---|---|---|---|---|
| Iteration number K | Training set | Validation set | Training set | Validation set |
| trained with original training set (K=0) | 99.43 | 89.16 | 100 | 93.75 |
| K=1 | 99.12 | 89.58 | 99.68 | 93.75 |
| K=2 | 99.63 | 89.17 | 100 | 95. 00 |
| K=3 | 99.71 | **92.81** | 100 | **96.25** |
| K=4 | 99.61 | 92.14 | 100 | 96.25 |

## 4 Conclusions

In this paper, we proposed a novel training strategy, namely reversed active learning (RAL) for the training of deep learning model. The CNN trained with proposed RAL was employed to address the challenge of breast cancer classification. The publicly available ICIAR2018 Breast Cancer Dataset (ICBD) was adopted to evaluate the performance of RAL-trained CNN. The experimental results show that the reversed active learning can significantly improve the CNN performance. The patch-based and slice-based accuracy of CNN on ICBD validation set increases to 92.81% and 96.25%, respectively.

### Acknowledgments

The work was supported by Natural Science Foundation of China under grands no. 61672357 and 61702339, the Science Foundation of Shenzhen under Grant No. JCYJ20160422144110140, and the China Postdoctoral Science Foundation under Grant No. 2017M622779.

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
