# OpenReview forum: "Active Learning for Breast Cancer Identification"
_MIDL.amsterdam/2018/Conference — Submitted to MIDL 2018_

### Review · AnonReviewer3 · 2018-05-04
**a neat idea with a simple trick that enables overall good results.**

**Rating:** 4
**Confidence:** 3

**Review:**

The paper presents a neat idea called reversed active learning (RAL)  to deal with background patches. The intuition is that even in an image labeled as malignant, there are many background patches that are normal-looking; hence they should be handled differently. Reversed active learning (RAL) is applied to filter out those background patches. Another simple trick is to use majority voting to reach a robust result.

The use of active learning in this context is quite novel. The experimental results are indeed better.

However,
- The writing should be improved. For example, the abstract does not really focus on the novelty. It needs proofreading.
- In the experiment, there is no comparison with other state-of-the-art method.

**Special Issue:**

Yes

---

### Review · AnonReviewer2 · 2018-05-08
**Interesting approach but not sure whether the method might over-fit**

**Rating:** 1
**Confidence:** 2

**Review:**

Authors present a “reverse” active learning approach for classification of histopathology images. The proposed method addresses the adverse effects of “mis-labeled” images on the training. First a CNN is trained on the entire set and images it is uncertain about are removed from the training set. This procedure is repeated for some iterations to yield the final classification.

Pros:
1. Article is clearly written.
2. The proposed method is easy to implement.
3. Empirical improvements are observed.
Cons:
1. The proposed method is most likely prone to over-fitting and this is not discussed. Given a set of images, the first CNN might be focusing on the wrong aspects of the images for prediction. Removing the images it is uncertain about will simply strengthen its attention and lead to stronger over-fitting. It is difficult to imagine how the proposed method can avoid this. As a result, it is not clear how the method can generalise.
2. I do not see how the method can automatically identify mislabeled images without additional ground-truth information.
3. I am not sure if the problem authors are addressing is really a problem in histopathology. In histopathology classification, a small part of the image can dictate the final class. Labels are often given image-wide and localisation is not performed while manual annotating. This is not necessarily mislabeling problem.
4. During slice-based classification, authors use majority voting to compute the final prediction. However, in these images, it is unclear whether majority of the image dictates the label. For instance, in Figure 3 in the article, the situation seems to be otherwise. Benign region in the image yield the final label for the slice even though some areas in image are normal.

In summary, authors take an interesting approach but the methodological details are confusing and possible over-fitting problems are not addressed.

**Special Issue:**

No

---

### Review · AnonReviewer1 · 2018-05-09
**The paper is not ready to be publish anywhere**

**Rating:** 1
**Confidence:** 3

**Review:**

Overall:
The paper proposes an active learning procedure to remove mislabeled patches (512x512) from large histopathology slides. In order to provide diagnosis a deep neural network is used. The problem is of high practical significance, however, it lacks a lot of important details. Moreover, the experiment mises many important details and, thus, it is hard to properly assess the presented idea.

Strengths:
+ Active learning approach is an important problem in both machine learning and medical imaging.
+ The breast cancer analysis using histopathology slides is of high practical significance.

Remarks:
* Major
- Some parts of the paper are hard to follow. I do not fully understand why mislabeled patches are removed from the dataset. Why is such process helpful? There is no motivation provided. Moreover, some explanations are very misleading. For instance, on page 3, the authors refer to Figure 3 where a refined dataset is presented. However, in this figure an example of a mislabeled patch is provided. Further, above Table 1, the authors state that: "(...) the network was continuously trained till the CNN achieved a satisfactory classification performance". How was the classification performance calculated? On training or validation data?
- Quality of Figure 4 and Figure 5 are too low.
- The description of the network architecture (Section 2.3) too laconic and it is hard to understand why such architecture was used. In fact, I would expect to see an experiment with and without DEN to see whether it is crucial in extracting useful features. It is missing in the experiments.
- I have huge doubts about what kind of slides were used. In Figure 6 it seems that these are of different resolution. If it is true then I wonder how a neural network (or any other model) can learn any useful features using the presented architecture?
- How was the hyperparameter search performed? Since the dataset was divided into 80% of training images and the remaining 20% was used for testing, I wonder whether the hyperparameters were tuned on test examples. If it is the case, then the results are completely unreliable.
- From the experiment it is extremely difficult to evaluate how useful is the proposed approach. First, it is doubtful whether a hyperparameter search was performed properly. Second, taking different values of K (number of iterations), the classification accuracy decreases and increases, e.g., for K=1 the patch-based ACA is 89.58, then for K=2 it decreases to 89.17 and eventually it increases to 92.81 for K=3. I find it counterintuitive and cannot think of a good explanation for that.

* Minor
- The LaTex style is incorrect. Fonts and spacing are wrong.
- It is worth to discuss relevant recent papers, for instance:
o active learning:
Gal, Y., Islam, R., & Ghahramani, Z. (2017). Deep bayesian active learning with image data. ICML 2017.
o Multiple Instance Learning for medical imaging:
Cheplygina, V., de Bruijne, M., & Pluim, J. P. (2018). Not-so-supervised: a survey of semi-supervised, multi-instance, and transfer learning in medical image analysis. arXiv preprint arXiv:1804.06353.
Ilse, M., Tomczak, J. M., & Welling, M. (2018). Attention-based deep multiple instance learning. arXiv preprint arXiv:1802.04712.
- The paper lacks a proper discussion of related work.
- The proposed approach completely disregards spatial dependencies among patches. It would be interesting to discuss possible extensions of the proposed approach regarding spatial dependencies.

**Special Issue:**

No

---

### Decision · Program_Chairs · 2018-05-15
**Paper3 Acceptance Decision**

Reject